# Optimization of an Analytical Protocol for the Extraction of Microplastics from Seafood Samples with Different Levels of Fat

**DOI:** 10.3390/molecules27165172

**Published:** 2022-08-13

**Authors:** Diogo M. Silva, C. Marisa R. Almeida, Francisco Guardiola, Sabrina M. Rodrigues, Sandra Ramos

**Affiliations:** 1ICBAS-Institute of Biomedical Sciences Abel Salazar-Porto University, Rua de Jorge Viterbo Ferreira nº 228, 4050-313 Porto, Portugal; 2CIIMAR-Interdisciplinary Centre of Marine and Environmental Research, University of Porto, Terminal de Cruzeiros do Porto de Leixões, Avenida General Norton de Matos, S/N, 4450-208 Matosinhos, Portugal; 3Chemistry and Biochemistry Department, Faculty of Science of University of Porto, Rua do Campo Alegre, 687, 4169-007 Porto, Portugal; 4Faculty of Biology, C. Campus Universitario, University of Murcia, 5, 30100 Murcia, Spain

**Keywords:** microplastics, canned seafood, fish, fat, methodology

## Abstract

Marine organisms are affected by the ubiquitous occurrence of microplastics (MPs) in the environment. Several protocols have been described to extract and quantify MPs in seafood, although their complex matrices, with high level of fat, can compromise the efficiency of MPs extraction. To solve this issue, the present study aimed to develop a detailed methodology suitable to process seafood samples with different levels of fat, namely fish and molluscs, from fresh and canned sources, including the immersive liquids from the cans. Sample digestion was tested using different solutions (10% KOH, 30% H_2_O_2_), temperatures (40 °C, 65 °C) and incubation times (24, 48, 72 h). For fat removal, three detergents (two laboratory surfactants and a commercial dish detergent) and 96% ethanol were tested, as well as the manual separation of fat. The methodology optimized in this study combined a digestion with 30% H_2_O_2_ at 65 °C, during 24 to 48 h, with a manual separation of the fat remaining after the digestion. All steps from the present methodology were tested in six types of polymers (PE-LD, PET, PE, AC, PS, and lycra), to investigate if these procedures altered the integrity of MPs. Results showed that the optimized methodology will allow for the efficient processing of complex seafood samples with different fat levels, without compromising MPs integrity (recoveries rate higher than 89% for all the polymers tested).

## 1. Introduction 

The global production of plastics has continually and dramatically increased, from 1.5 million tons in 1950 to approximately 370 million tons in 2020 [1]. Due to this growth and the lack of proper waste management over the years, there is a proliferation of plastics, and consequently microplastics (MPs), in the environment. MPs are small plastic particles, less than 5 mm in size, that can be intentionally manufactured for industrial uses (primary MPs) or originated from the degradation of larger plastics items (secondary MPs) [2]. According to recent studies, MPs are considered ubiquitous, and their occurrence in marine organisms has been reported, including plankton [3], fish and molluscs [4,5,6,7] as well as in processed seafood, such as canned products [8,9,10]. Due to the human consumption of a wide variety of seafood, consumers may be at risk of ingesting MPs, originating a growing concern regarding the exposure to these particles and their potential harm to human health [11,12].

Seafood is an important source of essential nutrients, especially high-quality proteins (including all of the essential amino acids), unsaturated fatty-acids (including Omega-3), fat-soluble vitamins, and minerals, with nutritional and physiological importance for human health [13,14]. In addition, the consumption of seafood as a major source of proteins is expected to increase as a consequence of the global population growth, with the demand for fish expected to almost double in 2050 [15,16]. Despite being one of the most commercialized food products, seafood is also highly perishable due to their biological composition, and several preservation methods, such as salting, freezing or canning, are traditionally used to control and ensure the quality of the products [17,18]. Canned seafood is an important food source and one of the most relevant ways of fish preservation, assuring the nutritional quality of food for long-term storages without the need for temperature control [19]. About 17 million tons of the world fishery production, corresponding to approximately 10% of the total 178 million tons, are preserved by canning [20,21,22]. There is a wide variety of canned seafood (e.g., fish, molluscs, crustaceans), immersed in different edible liquids (e.g., sunflower oil, olive oil, tomato sauce, garlic sauce), each one with specific ingredients and nutritional composition (e.g., fat, carbohydrates, proteins). In general, consumers eat these products directly from the can without any additional cleaning process. Therefore, from a dietary and health perspective, a better understanding on the potential occurrence of MPs is crucial [10]. Additionally, as canned products can be consumed with ingestion of both food and the respective filling liquids, both edible fractions should be included in the studies. 

However, seafood samples, particularly from canned products, have specific characteristics that can compromising the methodologies usually applied for extractions of MPs from water [23,24], sediment [23,25] and even other biota [26,27] samples. The presence of high levels of fat can be an issue, because the digestion of the organic matter, the filtration, and the subsequent MPs observation and retrieving can be very challenging or even compromised. From all the literature concerning MPs occurrence in a wide diversity of seafood, to the best of our knowledge only one paper focused on the particularities and adversities of MPs recovery from complex lipid-rich samples [28], and only four papers assessed MPs contamination in samples of canned seafood [8,9,10,29]. So, more research on the topic is needed taking in consideration the variability of seafood available. 

To solve this methodological issue, the present study aimed to develop a detailed protocol to extract and characterize MPs suitable to be applied to different matrices of seafood, such as fish and molluscs from fresh and canned sources, with different levels of fat. 

## 2. Methodology

The proposed methodology for MPs recovery from seafood samples comprises the following stages: organic content digestion, fat post-digestion removal, filtration and analysis of retrieved MPs. A summarized schematic representation is shown in Figure 1 with optimization options tested. Before starting the laboratory procedures, rigorous and proper measures were adopted to avoid external contaminations, as explained in Section 2.3.

### 2.1. Laboratorial Procedures

#### 2.1.1. Sample Digestion

Six tests were conducted to optimize the digestion of a variety of seafood samples with different levels of fat (Table 1, Figure 2). 

The efficiency of each digestion test was assessed through visual inspection and expressed as the percentage of digestion of the sample: 100%-total digestion of the sample, with no visible remains at the end of the test; 95%-digestion of almost the totality of the sample, with few visible remains at the end of the test; 75%-partial digestion of the sample, with visible remains at the end of the test; 50%-digestion of approximately half of the sample, with visible remains at the end of the test.

#### 2.1.2. Fat Post-Digestion Treatment 

After digestion, samples follow to the next step that is filtration. Nevertheless, for samples with high levels of fat, filtration was not possible because the filters clogged easily with the fat. Hence, five tests were performed to dissolve or remove the remaining fat in digested samples (Table 2, Figure 3).

The efficiency of each test of fat elimination was assessed through visual inspection, and expressed as the percentage of fat removed from the sample: 100%-total removal of the fat from the sample, with no visible remains at the end of the test; 75%-partial removal of the fat from the sample, with visible remains at the end of the test; 50%-digestion of approximately half of the fat from the sample, with visible remains at the end of the test; 25%-digestion of a small part of the fat from the sample, with visible remains at the end of the test. 

#### 2.1.3. Filtration and Analysis of MPs

Solutions obtained from the digestion of samples with low levels of fat were directly subjected to vacuum filtration using cellulose nitrate membrane filters (pore size 0.45 µM, 47 mm diameter). The other samples, after fat removal, were also filtered using the same conditions.

After filtration, filters were stored in glass Petri dishes, air-dried at room temperature for two days, and observed in a stereomicroscope (Leica EZ04). Microplastics were classified according to the type (e.g., fiber, fragment, film), color and size, and photographed. 

### 2.2. Microplastics Integrity Test

After the development of the optimized methodology for the extraction and characterization of MPs from seafood samples with different levels of fat, the full protocol was applied to known and common types of plastic polymers to evaluate possible effects in the integrity of MPs. The most common types of plastic polymers observed in marine litter (Plastics Europe, 2021) were tested, namely: low-density polyethylene (PE-LD), polyethylene terephthalate (PET), polyethylene (PE), acrylic (AC), polystyrene (PS), and lycra. In the laboratory, all MPs were manually produced from common items (e.g., fishing line, bottle of water, plastic bags, clothes), and sieved through a 5 mm mesh to discard particles larger than 5 mm. An initial mass of 0.002 g of each type of polymer was used to test MPs integrity with all the laboratorial procedures optimized in this study (more in Section 4). 

### 2.3. Contamination Control

Seafood sampling was made in a laboratory with restricted access. A clean laboratory coat (gray, 100% cotton) and single-use nitrile gloves were used during the entire processing. All working surfaces were thoroughly cleaned with 70% EtOH, and all glassware and metallic tools were rinsed with filtered deionised water and 70% EtOH before use. The outer part of the fish and seafood cans were also rinsed to eliminate any potential particles attached to the surface. Liquid solutions were regularly observed under a stereomicroscope to assess the occurrence of MPs contamination; if necessary, solutions were filtered through sterile cellulose nitrate membrane filters (pore size 0.45 µM) prior to use. Procedural blanks (a glass Petri dish with filtered deionised water) were left exposed in the cabinet of the laboratory where all steps were performed to capture any possible airborne contamination, and immediately observed at the stereomicroscope.

## 3. Results

### 3.1. Sample Digestion

The efficiency of the sample digestion tests was assessed through visual inspection, and results were provided in percentage of organic matter digested as described in the experimental section (Table 3).

**Table 3 molecules-27-05172-t003:** Organic matter digestion rate (%) of samples from each test (after 24, 48 and 72 h of incubation) with specific observations.

		Digestion Rate (%)			
Test	Sample	24 H	48 H	72 H	Observations
A1	Fresh low-fat fish muscle (n = 2)	75	95	95	Incomplete digestion; orange/reddish and viscous digestion liquid (Figure 4A); * bone remains (Figure 4B)
	Fresh low-fat fish GIT (n = 2)	50	75	95	
	Fresh low-fat fish Gill (n = 2)	50	75	95 *	
A2	Fresh low-fat fish muscle (n = 2)	95	95	95	Incomplete digestion; orange/reddish and viscous liquid; * bone remains
	Fresh low-fat fish GIT (n = 2)	75	95	95	
	Fresh low-fat fish Gill (n = 2)	75	95	95 *	
B1	Fresh low-fat fish muscle (n = 2)	50	100	100	Complete digestion; translucent and fluid liquid (Figure 4A);
	Fresh low-fat fish GIT (n = 2)	50	100	100	
	Fresh low-fat fish Gill (n = 2)	50	95	95	Incomplete digestion: bone remains; translucent and fluid liquid
B2	Fresh low-fat fish muscle (n = 2)	50	100	100	Complete digestion; translucent and fluid liquid
	Fresh low-fat fish GIT (n = 2)	50	100	100	
	Fresh low-fat fish Gill (n = 2)	50	95 *	95 *	Incomplete digestion: * bone remains; translucent and fluid liquid
C1	Fresh high-fat fish muscle (n = 4)	50	100	-	Complete digestion; fat droplets (Figure 4C)
	Fresh high-fat fish GIT (n = 4)	50	100	-	
	Fresh high-fat fish Gill (n = 4)	50	95	-	Incomplete digestion: bone remains
	Canned fish (n = 4)	50	95	-	Incomplete digestion: bone remains; fat droplets
	Canned mollusc (n = 4)	50	100	-	Complete digestion; fat droplets
	Canned oil (n = 4)	100	100	-	Complete digestion; layer of fat (Figure 4D)
	Canned sauce (n = 4)	100	100	-	
C2	Fresh high-fat fish muscle (n = 4)	50	100	-	Complete digestion; smaller fat droplets
	Fresh high-fat fish GIT (n = 4)	50	100	-	
	Fresh high-fat fish Gill (n = 4)	50	95	-	Incomplete digestion: bone remains
	Canned fish (n = 4)	50	95	-	Incomplete digestion: bone remains; smaller fat droplets
	Canned mollusc (n = 4)	50	100	-	Complete sample digestion; smaller fat droplets
	Canned oil (n = 4)	100	100	-	Complete sample digestion; smaller layer of fat
	Canned sauce (n = 4)	100	100	-	

**Figure 4 molecules-27-05172-f004:**
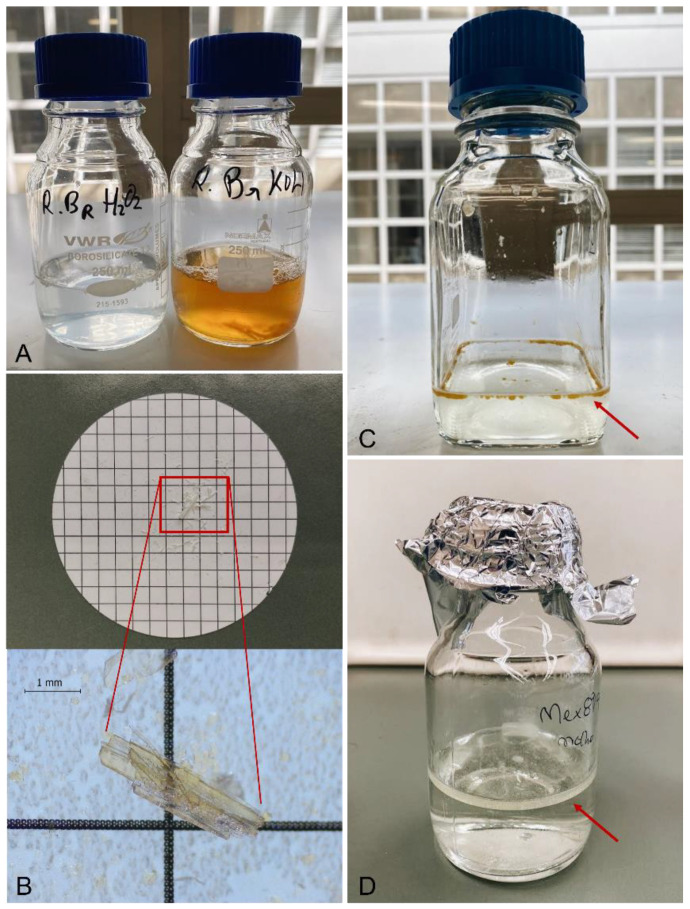
(**A**): Digestion of fresh fish gills with 30% H_2_O_2_ (left bottle, translucent and fluid liquid), and 10% KOH solution (right bottle, orange and viscous liquid); (**B**): Cellulose nitrate filter with undigested bony remains from a fresh fish gills sample; (**C**): Presence of small droplets of fat (red arrow) from the digestion of canned mussels with 30% H_2_O_2_; (**D**): Presence of a layer of fat (red arrow) from the digestion of the immersing liquid (escabeche sauce) of canned mussels with 30% H_2_O_2_.

Tests A1 and A2 (10% KOH at 40 °C and 65 °C, respectively) presented an incomplete digestion of the organic matter during the entire experiment, where a maximum digestion rate of 95% was observed for all the samples after 72 h of incubation. The final solution obtained from the digestion presented a viscous consistency, with an orange to reddish coloration for all the samples tested. In the fresh fish gills samples, bony remains from the branchial arches persisted undigested.

Regarding tests B1 and B2 (30% H_2_O_2_ at 40 °C and 65 °C, respectively), a complete digestion of the organic matter (100%) was achieved for all the samples at the incubation times of 48 and 72 h, except for the fresh fish gills samples. For the later, an incomplete digestion rate of 95% was obtained for the two incubation periods (48 and 72 h), with undigested bony remains from the branchial arches of the gills.

Concerning tests C1 (30% H_2_O_2_, at 65 °C) and C2 (30% H_2_O_2_, at 65 °C, with a pre-treatment overnight at 40 °C), with an incubation period of 48 h, a complete digestion (100%) of the organic matter was generally achieved. An exception was observed for the samples presenting bony structures (the branchial arches from the fresh fish gills and the bone from the canned fish), where a digestion rate of 95% was observed, with undigested remains. For all the samples tested in C1 and C2, the digestion of the organic matter originated the development of fat. For samples with lower levels of fat small lipidic droplets were formed. In samples with high levels of fat, a lipidic layer was formed at the surface of the digested solution. Comparing the fat portion at the end of the digestion period for tests C1 and C2, there was a visible decrease in the levels of fat in test C2, with all the samples presenting smaller droplets or layers of fat-a very important feature for samples with high levels of fat. The visual inspection of the samples after 24 h of incubation demonstrated a complete digestion of the organic matter from the canned oil and canned sauce samples (100%). 

So, the protocol chosen for the sample digestion was the C2: pre-treatment of seafood samples in a laboratory oven at 40 °C overnight, and sample digestion carried out with 30% H_2_O_2_ (with a volume capable of covering the entire sample), incubated in a laboratory oven at 65 °C for 24 to 48 h (the incubation time differs according to the characteristics of the sample).

### 3.2. Fat Post-Digestion Treatment

The efficiency of the tests for fat post-digestion treatment was assessed through visual estimation, and results were provided in percentage of fat removal (Table 4).

The three tests using surfactant compounds (D–F) presented similar results, with all of them showing an incomplete fat dissolution in all the tested samples. For samples with higher levels of fat (canned oil and canned sauce), the percentage of fat removal was comparably lower (25%) than from the other samples (50%). Further, tests D, E and F also led to the formation of bubbles (plus foam, in the case of test F), which limited the sample visualization and the following steps of the protocol.

Regarding test G (96% EtOH), an incomplete fat dissolution was observed for all the samples. Again, samples with higher levels of fat (canned oil and canned sauce) presented a lower rate of fat removal (25%) when compared to fresh high-fat fish samples (75% of fat removal). For these last samples, and in opposition to the previous tests with surfactants, the fat dissolution with 96% EtOH allowed to subsequently filtrate the samples. However, since fat removal was incomplete, the filters easily clogged, making the filtration a very slow and time-consuming process. 

Test H achieved the best results, presenting a complete fat removal (100%) for all the studied samples. Manually, and with the help of a steel spoon, fat (frequently occurring in the surface of the digested solution) was collected and transferred to a clean Petri dish, and immediately observed under a stereomicroscope for the presence of MPs. 

So, the protocol chosen for the fat post-digestion treatment was the H: manual recovery of the remaining fat, with immediate observation under a stereomicroscope.

## 4. Microplastics Integrity Test

Polymers were placed in individualized glass flasks and incubated with 50 mL of 30% H_2_O_2_, at 65 °C for 48 h. After digestion, solutions containing the plastic polymers were vacuum filtered using cellulose nitrate filter membranes (pore size 0.45 µM). After 48 h of drying at room temperature, filters were weighed in order to have the final mass and recovery rate of each polymer. Filters were observed under a stereomicroscope to conclude about the polymers’ morphological characteristics after the digestion incubation (Table 5). 

Results showed that recovery rates ranged from a minimum of 89.0% (for lycra) to a maximum of 98.2% (for PE-LD), with an average recovery rate for all the polymers of 96.6%, confirming the efficiency of the proposed methodology for the recovering of MPs from seafood samples. At the end of the protocol, only PE-LD polymers (recovery rate of 98.2%) presented differences from their original characteristics, specifically color, which changed from blue to white. Tests with AC showed a recovery rate above 100% (recovery rate of 102%) but, since the initial and final mass values are very similar, it can be explained as the result of an incomplete drying of the filter before the final weighing.

## 5. Conclusions

The present work allowed for the optimization of an analytical protocol for the extraction of MPs from seafood samples with different levels of fat, with the following considerations: 

Pre-treatment of seafood samples in a laboratory oven at 40 °C overnight;

Sample digestion with 30% H_2_O_2_ (with a volume capable of covering the entire sample), incubated in a laboratory oven at 65 °C, during 24 to 48 h (the incubation time may vary according to the characteristics of the sample);

Fat post-digestion treatment with manual recovery of the remaining fat, with immediate observation under a stereomicroscope;

Vacuum filtration of the digested sample (without fat) through cellulose nitrate filter membranes. After performing all protocol steps, a Fourier Transform Infrared Spectroscopy (FTIR) analysis, or an equivalent technique, is necessary to identify the polymers of the collected MPs.

The optimized protocol showed high recovery rates and was also efficient in guarantying the integrity of MPs for the majority of plastic polymers tested. Hence, this protocol is suitable to retrieve MPs properly and efficiently from seafood samples with distinct levels of fat, including fresh and canned seafood. The methodology was already successfully applied in several seafood samples (namely fresh sardines, fresh mussels, and a variety of canned samples such as sardine, tuna, mussels, octopus and chub mackerel), allowing the observation and extraction of MPs for all the sampled species (Figure 5). 

## Figures and Tables

**Figure 1 molecules-27-05172-f001:**
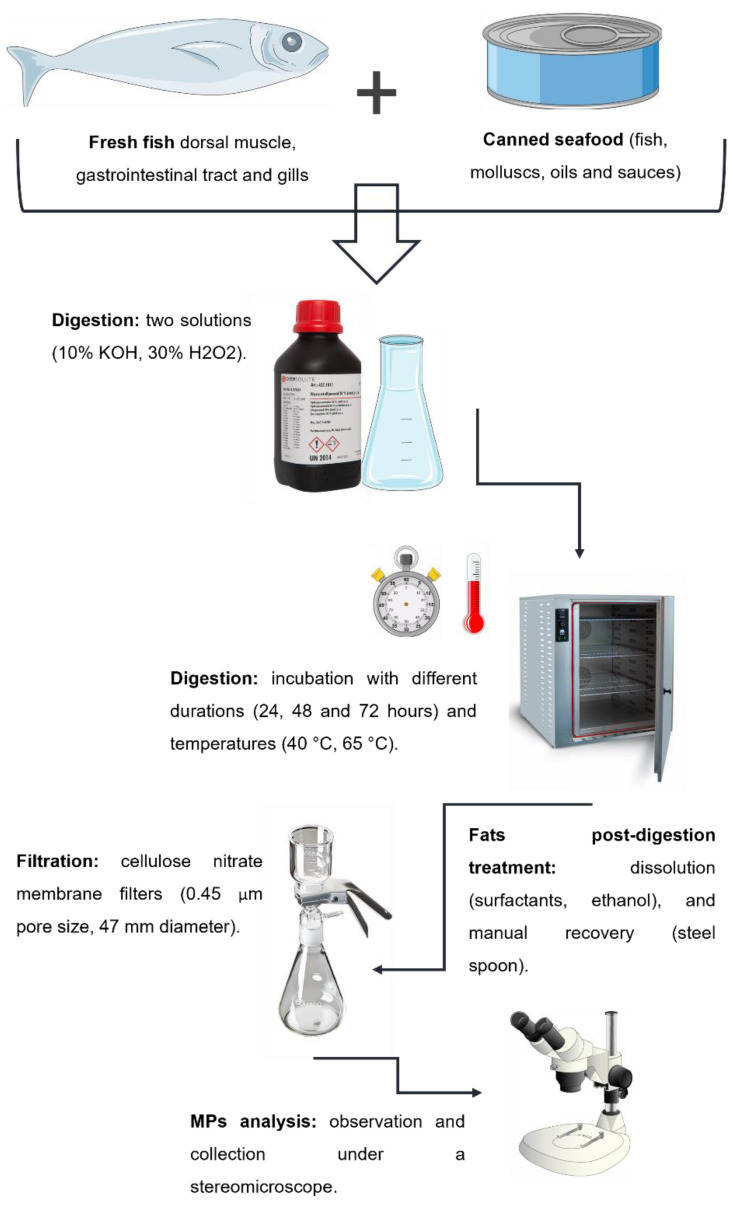
Schematic representation of the different stages of the proposed methodology, with optimization options tested, for MPs extraction from seafood samples with different levels of fat.

**Figure 2 molecules-27-05172-f002:**
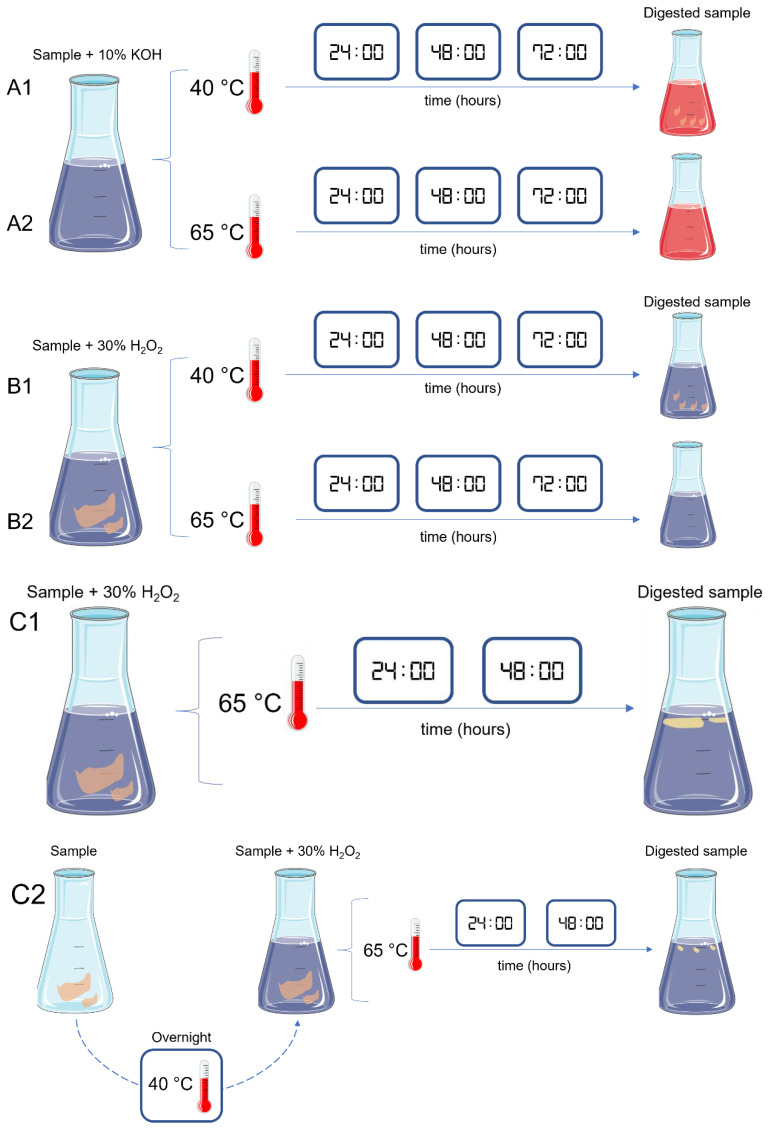
Schematic representation of the different tests (A1, A2, B1, B2, C1 and C2) performed for the digestion of seafood samples with different levels of fat.

**Figure 3 molecules-27-05172-f003:**
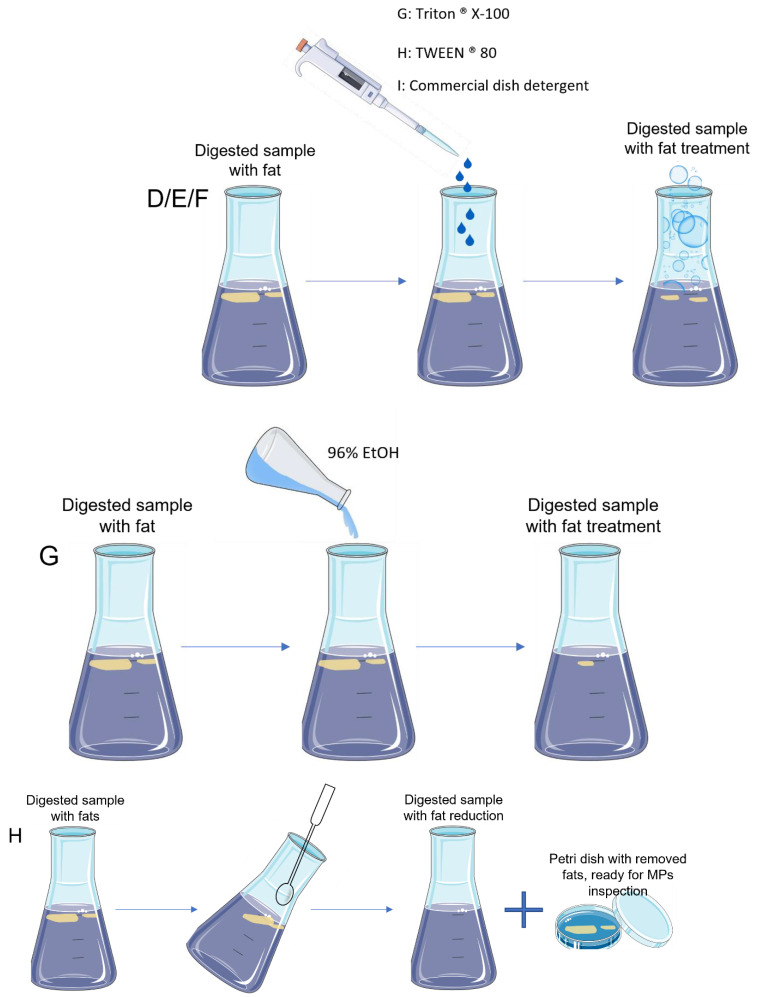
Schematic representation of the different tests (D, E, F, G and H) performed to dissolve or remove fat from the digested seafood samples.

**Figure 5 molecules-27-05172-f005:**
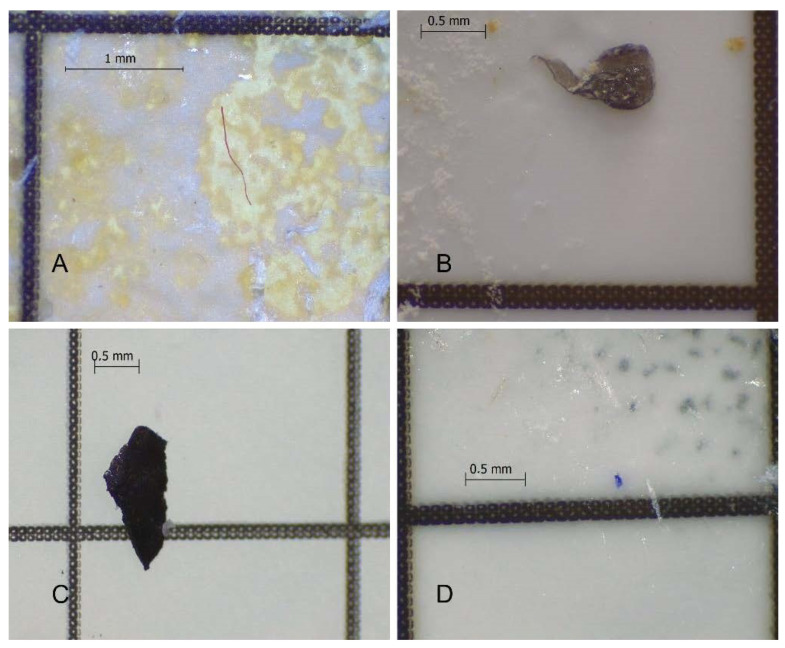
Examples of microplastics recovered with the optimized methodology. (**A**): red fiber from canned chub mackerel; (**B**): black film from fresh mussel; (**C**): black fragment from canned sardine; (**D**): blue film from canned sardine liquid.

**Table 1 molecules-27-05172-t001:** General information about the different tests (A1, A2, B1, B2, C1 and C2) performed for the digestion of seafood samples with different levels of fat.

Test	Sample Pre-Treatment	Solution	Volume ^1^ (mL)	Duration (Hours)	Temperature (°C)	Sample ^2^
A1	-	10% KOH	50–70	72	40	Fresh low-fat fish (muscle, GIT and gills)
A2	-	10% KOH	50–70	72	65	
B1	-	30% H_2_O_2_	50–70	72	40	Fresh low-fat fish (muscle, GIT and gills)
B2	-	30% H_2_O_2_	50–70	72	65	
C1	-	30% H_2_O_2_	50–70	48	65	Fresh high-fat fish (muscle, GIT and gills); canned fish, canned mollusc, canned oil, canned sauce
C2	Overnight 40 °C	30% H_2_O_2_	50–70	48	65	

^1^ Volume defined as the minimum quantity (in mL) capable of covering the entire sample to be digested. ^2^ Fresh low-fat fish samples (dorsal muscle, GIT and gills) obtained from *Trisopterus luscus* and *Dicentrarchus labrax*. Fresh high-fat fish samples (dorsal muscle, GIT and gills) obtained from *Sardina pilchardus*. Canned samples from tuna, sardine, octopus and mussels, immersed in sunflower oil, tomato sauce and “escabeche” sauce.

**Table 2 molecules-27-05172-t002:** General information about the different tests (D, E, F, G and H) performed to dissolve or remove fat from the digested samples.

Test	Fat Treatment	Volume	Sample
D	Triton^®^ X-100	2% of sample volume	Fresh high-fat fish (muscle and GIT); canned oil, canned sauce
E	TWEEN^®^ 80	2% of sample volume	Fresh high-fat fish (muscle and GIT); canned oil, canned sauce
F	Commercial dish detergent	2% of sample volume	Fresh high-fat fish (muscle and GIT); canned oil, canned sauce
G	96% EtOH	Same as sample volume (1:1 *v*/*v*)	Fresh high-fat fish (muscle and GIT); canned oil, canned sauce
H	Removed manually	-	Fresh high-fat fish muscle and GIT; canned oil, canned sauce

**Table 4 molecules-27-05172-t004:** Fat removal rate (%) of samples from each test, and specific notes.

Test	Sample	Fat Removal Rate (%)	Notes
D	Fresh high-fat fish muscle (n = 2)	50	Incomplete fat dissolution with formation of bubbles; difficult visualization
	Fresh high-fat fish GIT (n = 2)	50	
	Canned oil (n = 1)	25	
	Canned sauce (n = 1)	25	
E	Fresh high-fat fish muscle (n = 2)	50	Incomplete fat dissolution with formation of bubbles; difficult visualization
	Fresh high-fat fish GIT (n = 2)	50	
	Canned oil (n = 1)	25	
	Canned sauce (n = 1)	25	
F	Fresh high-fat fish muscle (n = 2)	50	Incomplete fat dissolution with formation of foam and bubbles; impossible visualization
	Fresh high-fat fish GIT (n = 2)	50	
	Canned oil (n = 1)	25	
	Canned sauce (n = 1)	25	
G	Fresh high-fat fish muscle (n = 2)	75	Incomplete fat dissolution; slow subsequent filtration
	Fresh high-fat fish GIT (n = 2)	75	
	Canned oil (n = 1)	25	Incomplete fat dissolution; impossible subsequent filtration
	Canned sauce (n = 1)	25	
H	Fresh high-fat fish muscle (n = 2)	100	Complete fat removal
	Fresh high-fat fish GIT (n = 2)	100	
	Canned oil (n = 1)	100	
	Canned sauce (n = 1)	100	

**Table 5 molecules-27-05172-t005:** Information about MPs polymers used in the integrity test. Recovery rate (%) = (Initial mass/final mass) * 100.

MPs Polymer	Initial Mass (g)	Final Mass (g)	Recovery Rate (%)	Observations
PE-LD	0.00217	0.00213	98.2	Color changed
PET	0.00232	0.00225	96.9	Without changes
PE	0.00203	0.00195	97.5	Without changes
AC	0.00200	0.00204	102.0	Without changes
PS	0.00220	0.00211	95.9	Without changes
Lycra	0.00200	0.00178	89.0	Without changes

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
