# Peer review of "Optimization of an Analytical Protocol for the Extraction of Microplastics from Seafood Samples with Different Levels of Fat"

_molecules, 2022, doi:10.3390/molecules27165172_

Round 1

Reviewer 1 Report

The manuscript on "Optimization of an analytical protocol for the extraction of microplastics from seafood samples with different levels of fat" represents an extensive study of a very pertinent problem in seafood causing significant quality issues. The authors have designed the experiments effectively to address the problem of fat in seafood causing hindrance in the extraction of microplastics from the samples.

The manuscript is worth publication after addressing the following:

1. English language editing

2. Schematic representation of methods are to be given only for those designed by the authors, and not for routine protocols

Author Response

1. English language editing - Ok, thanks for the advice. The english will be revised.

2. Schematic representation of methods are to be given only for those designed by the authors, and not for routine protocols - Ok, thanks. All the tests represented with schemes were designed by me and the following authors, some of them through the combination of different protocols already published (example: the digestion solution of a protocol, with the incubation time of another, with a lower temperature than the usually published).

Reviewer 2 Report

The manuscript “Optimization of an analytical protocol for the extraction of microplastics from seafood samples with different levels of fat.” by Silva et al. is organized very well. The authors optimized the protocol for the extraction of microplastics from different seafood products (low and high in fat). They performed 6 different tests for the seafood samples' digestion and 5 different tests for fat removal from the digested samples. The result is a promising protocol for the extraction of microplastics from seafood samples. Besides, the authors performed the integrity test with the most common types of plastic polymers found in the marine litter to test the optimized protocol. The paper is written very well and the graphics are quite vivid. This protocol will encourage more research in the field. I would suggest considering this manuscript for publication after correcting a few details:

The title ends with a dot – is it a mistake?

2.2.3. Please remove FTIR from this section, there are no results of FTIR. You can put to the conclusion that this analysis is necessary to identify polymers.

Line 249 – change um to µm

References should be numbered in the text according to the Journal’s instructions.

Author Response

The title ends with a dot – is it a mistake? - Yes, dot was removed. Thanks.

2.2.3. Please remove FTIR from this section, there are no results of FTIR. You can put to the conclusion that this analysis is necessary to identify polymers. - Ok, changed. Thanks

Line 249 – change um to µm - Ok, corrected! Thanks.

References should be numbered in the text according to the Journal’s instructions - Ok, corrected. Thanks
